# Protective Effect of Neutral Electrolyzed Saline on Gentamicin-Induced Nephrotoxicity: Evaluation of Histopathologic Parameters in a Murine Model

**DOI:** 10.3390/medicina59020397

**Published:** 2023-02-17

**Authors:** Nomely S. Aurelien-Cabezas, Brenda A. Paz-Michel, Ivan Jacinto-Cortes, Osiris G. Delgado-Enciso, Daniel A. Montes-Galindo, Ariana Cabrera-Licona, Sergio A. Zaizar-Fregoso, Juan Paz-Garcia, Gabriel Ceja-Espiritu, Valery Melnikov, Jose Guzman-Esquivel, Iram P. Rodriguez-Sanchez, Margarita L. Martinez-Fierro, Ivan Delgado-Enciso

**Affiliations:** 1School of Medicine, University of Colima, Colima 28040, Mexico; 2Department of Research, Esteripharma SA de CV, Atlacomulco 50450, Mexico; 3Cancerology State Institute, Colima State Health Services, Colima 28085, Mexico; 4School of Chemical Sciences, University of Colima, Coquimatlan 28400, Mexico; 5Union Hospital Center, Villa de Álvarez, Colima 28970, Mexico; 6Clinical Epidemiology Research Unit, Mexican Institute of Social Security Institute, Villa de Álvarez 28984, Mexico; 7Molecular and Structural Physiology Laboratory, School of Biological Sciences, Universidad Autónoma de Nuevo León, San Nicolás de los Garza 66455, Mexico; 8Molecular Medicine Laboratory, Unidad de Medicina Humana y Ciencias de la Salud, Universidad Autónoma de Zacatecas, Zacatecas 98160, Mexico

**Keywords:** anti-inflammatory agents, gentamicin, hypochlorous acid, kidney, reactive oxygen species, toxicity

## Abstract

*Background and Objectives*: Gentamicin (GM) is a nephrotoxic aminoglycoside. Neutral electrolyzed saline (SES) is a compound with anti-inflammatory, antioxidant, and immunomodulatory properties. The objective of the present study was to evaluate whether kidney damage by GM can be prevented and/or reversed through the administration of SES. *Materials and Methods*: The study was carried out as a prospective, single-blind, five-arm, parallel-group, randomized, preclinical trial. The nephrotoxicity model was established in male BALB/c mice by administering GM at a dose of 100 mg/kg/day intraperitoneally for 30 days, concomitantly administering (+) SES or placebo (physiologic saline solution), and then administering SES for another 30 days after the initial 30 days of GM plus SES or placebo. At the end of the test, the mice were euthanized, and renal tissues were evaluated histopathologically. *Results*: The GM + placebo group showed significant tubular injury, interstitial fibrosis, and increased interstitial infiltrate of inflammatory cells compared with the group without GM. Tubular injury and interstitial fibrosis were lower in the groups that received concomitant GM + SES compared with the GM + placebo group. SES administration for 30 days after the GM administration periods (GM + placebo and GM + SES for 30 days) did not reduce nephrotoxicity. *Conclusions*: Intraperitoneal administration of SES prevents gentamicin-induced histologic nephrotoxicity when administered concomitantly, but it cannot reverse the damage when administered later.

## 1. Introduction

Drug-related nephrotoxicity is a cause of acute kidney injury (AKI) in patients hospitalized in intensive care units (ICUs) [1]. Six months after presenting AKI induced by nephrotoxic drugs, 70% of patients present evidence of residual kidney damage: reduced glomerular filtration rate, hyperfiltration, proteinuria, or hypertension [2]. Aminoglycosides are a group of antibiotics still widely used at the hospital level, due to their low cost and broad bactericidal effects [3]. Cumulative aminoglycoside use is associated with long-term renal impairment [4]. Gentamicin (GM), a drug belonging to the aminoglycoside family, is known to be nephrotoxic [5,6]. Most studies have reported rates of nephrotoxicity between 8 and 26%, and it is estimated that 21.4% of pediatric patients admitted to ICUs receive GM [7].

In preclinical studies, gentamicin-induced nephrotoxicity is characterized by histologic changes, including tubular injury, tubular atrophy, and interstitial fibrosis [8,9,10,11,12]. Such effects are known to be triggered by the nitrosative and oxidative stress induced by the aminoglycoside [13,14].The administration of compounds with anti-inflammatory and antioxidant activity to reduce GM-induced nephrotoxicity has been effective [15,16], albeit their clinical efficacy is inconclusive [17].

Neutral electrolyzed saline (SES) is a neutral-pH solution that has demonstrated anti-inflammatory and immunomodulatory properties [18], as well as antiseptic activity [19,20]. SES is produced from a saline solution of sodium chloride and is activated by a controlled electrolysis process, which produces reactive species of chlorine and oxygen (ROS). The ROS that are generated are oxidizing chlorine species, such as hypochlorous acid (HOCl), and oxidizing oxygen species, such as hydrogen peroxide (H_2_O_2_). In addition, molecular hydrogen (H_2_) is generated during the process (pat. no. MX330845B).

In vitro, activated saline solution participates in an anti-inflammatory mechanism at the cellular level, inhibiting the secretion of TNF-α, IL-6, and MIP-1 [18]. Furthermore, SES has been reported to have an immunomodulatory and anti-inflammatory effect capable of increasing the number of total platelets and leukocytes in patients with COVID-19 [21]. Molecular hydrogen, an SES compound, has also been reported to have a pulmonary antifibrotic effect [22]. On the other hand, electrolyte-reduced water, with an alkaline pH and a significantly higher concentration of H2 than SES, has been used in hemodialysis as a dialysis solution, reducing oxidative stress and the proinflammatory cytokine profile [23]. However, the latter study did not analyze whether molecular hydrogen prevented histopathologic alterations [23]. It is also well-established in the literature that SES can improve wound healing, not only because of its benefits as an antiseptic, but also due to its effects as an anti-inflammatory and cell-proliferation substance, through the mediation of oxidative and inflammatory responses [24,25,26]. This suggests that, to some extent and according to its components, SES or any of its components can protect against nephrotoxic agents. However, whether its potential beneficial effect is only preventive or whether it is also therapeutic (capable of reversing damage) has not been proven histologically, nor has it been evaluated.

Based on these premises, the present study aims to analyze the prophylactic and/or therapeutic effect of neutral electrolyzed saline in a murine model of gentamicin-induced nephrotoxicity.

## 2. Materials and Methods

### 2.1. Study Design

The study was carried out as a prospective, single-blind, 5-arm, parallel-group, randomized, preclinical trial (see Figure 1), according to the “ARRIVE Essential 10” guidelines for Animal Research [27].

### 2.2. Sample Size

The sample size was calculated using the formula based on incidence assessment [28], as previously reported [29]. The minimum number required to establish comparisons was nine animals per group. In this study, 53 mice were randomly divided into five groups.

### 2.3. Animals, Inclusion Criteria, and Randomization

Inclusion criteria were male BALB/c mice (Envigo, Coyoacan, Mexico City, Mexico) aged 12–14 weeks and weighing 20–30 g. Elimination criteria were any experimental subject that died during the study period or that developed any other type of illness. The animals were kept under suitable conditions at 21 ± 2 °C in a 12 h light/dark cycle, with food and water provided ad libitum. As previously reported, the mice were fed a high-fat diet to promote a renal and systemic inflammatory environment [30,31,32]. The mice were kept in cages, with a maximum of 5 mice per group. Fifty-three mice were divided into five parallel groups using a randomized block design [33] (see Figure 1).

### 2.4. Ethics

This study was approved by the Research Ethics Committee of the Colima State Institute of Cancerology (Colima, Colima, Mexico) (protocol no. CIIECAN/06/19). The animals were handled following institutional guidelines and the official Mexican norm that regulates the use of laboratory animals (NOM-062-ZOO-1999) [34], in addition to the Guide for the Care and Use of Laboratory Animals prepared by the National Academy of Sciences of the USA (2011) [35]. Furthermore, all the animals were euthanized by decapitation according to the American Veterinary Medical Association Guidelines for the Euthanasia of Animals, 2020 Edition [36].

### 2.5. Reagents, Animal Model, and Intervention

The gentamicin used was Genkova^®^ (SON’S, SA de CV, Puebla, Puebla, Mexico). Esteripharma provided a neutral electrolyzed saline solution (Atlacomulco, State of Mexico, Mexico) of parenteral-application quality (Homestec) [37], which was SES neutral pH, 0.0020% active chlorine^®^ [37]. Physiologic saline solution for intravenous application was used as the placebo.

The nephrotoxicity model was established in male BALB/c mice by administering GM at a dose of 100 mg/kg/day for 30 days intraperitoneally (IP), administering SES or placebo (150 µL/mice/day, IP) concomitantly, and later administering SES or placebo for another 30 days. Five groups were formed: (1) GM + SES-after placebo (to assess the protective effect of SES); (2) GM + placebo-after SES (to assess whether SES reversed the damage through its application after the GM-application period); (3) GM + SES-after SES; (4) GM + placebo-after placebo (to assess the maximum kidney damage caused by GM); and (5) Placebo-after placebo (reference of the absence of damage caused by GM). After the completion of the experiment, the mice were euthanized by decapitation (day 61). The dose of SES administered per day in the mouse model was equivalent to that given to human patients intravenously in a previous clinical trial, in which an 80 mL dose of SES per day was shown to have an anti-inflammatory and immunomodulatory effect on patients [21]. Using a calculator that estimates the dose scale between species of different body weights via exponential allometry (http://clymer.altervista.org/minor/allometry.html; accessed 2 February 2023) [38], a dose of 150 µL/day for the mice was found to be a dose equivalent to 80 mL/day in humans. This calculation considered an exponent of 0.80, as well as an average weight of mice and humans of 25 g and 70 Kg, respectively. The concomitant GM + SES-application period was 30 days. Blood samples were obtained afterward, and the serum was separated for IL-6, creatinine, and urea quantification. A Mouse IL-6 ELISA Kit RAB0308 (Sigma-Aldrich, Saint Louis, MO, USA) was used according to the manufacturer’s instructions, and the analyses were performed in triplicate. Serum creatinine and urea were determined using an automatic biochemical analyzer (Cobas c111, Roche^®^, Miguel Hidalgo, Mexico City, Mexico). The kidneys were simultaneously dissected for histologic processing. The researcher who performed the histopathologic assessment was blinded.

In the present study, GM was administered for 30 days because that application period is capable of causing permanent destruction of some of the tubules that reflect fibrosis [39,40]. The recovery period after the final GM application was 30 days, after which the mice were euthanized for histologic analysis. That time period was chosen because acute histologic changes have been reported to be fully reversed between 3 and 30 days after the cessation of GM administration, and so changes that remain after 30 days can be considered chronic and permanent [39,41]. It is important to point out that nephrotoxicity from administering GM for up to 30 days, despite being able to cause histologic lesions, does not alter blood urea nitrogen or creatinine because of glomerular-filtration-rate preservation due to a process of necrosis and cell regeneration [40]. Therefore, the mouse model analyzed in the present study represents early (preclinical) chronic kidney disease [42], considering that the common histopathologic features can predict renal failure [43].

### 2.6. Histopathologic Processing and Assessment

The tissues were fixed in a solution of 10% formaldehyde for at least 24 h. Renal tissue was embedded in paraffin. Tissue sections (5 µm) were prepared using a microtome and mounted on slides. Masson’s trichrome stain (Merck KGaA, Darmstadt, Hesse, Germany) was used to detect interstitial fibrosis. In addition, hematoxylin–eosin staining (Merck KGaA, Darmstadt, Hesse, Germany) was also carried out to detect tubular injury (blebbing of the apical membrane into the tubular lumen, cell fragments within the tubular lumen, flattening of the tubular epithelium, or loss of nuclei), tubular atrophy, and interstitial infiltrate of inflammatory cells [44]. Hematoxylin–eosin and Masson’s trichrome staining were performed according to standard procedures. Slices were evaluated via images captured with a Moticam 1080 digital camera (Motic China Group Co Ltd, Xiamen, Fujian, China) attached to a Moticam BA310E optical microscope (Motic China Group Co Ltd, Xiamen, Fujian, China) with 10× and 40× objectives. All images were captured under the same conditions of light and exposure.

Renal histopathologic lesions were calculated as the percentage of the total area observed under the microscope [45]. The total area of the renal tissue cut was considered 100%, and the percentage of the cortical area affected by tubular injury, interstitial fibrosis, tubular atrophy, and interstitial infiltration of inflammatory cells was quantified. Additionally, the total percentage of the renal cortex tissue altered histopathologically was calculated and was the result of the sum of the percentages of tubular injury, tubular atrophy, fibrosis, and inflammatory infiltrate in the renal cortex. Tubular injury was defined as the flattening of the tubular epithelium with calcified or noncalcified cellular fragments within their lumens, blebbing of the apical membrane into the tubular lumen, or loss of nuclei [44]. Interstitial fibrosis was defined as increased extracellular matrix separating tubules in the cortical area [46], demonstrated as the blue-stained areas on Masson’s trichrome stains [47]. Tubular atrophy was defined by thick, irregular tubular basement membranes, with decreased diameters of tubules [46]. Interstitial infiltrate of inflammatory cells was defined as an excess of inflammatory cells within the cortical interstitium [46]. The evaluations were carried out blindly by two anatomopathologic experts.

### 2.7. Statistical Analysis

The data distributions were not normal (except for serum creatinine and urea values), according to the Kolmogorov–Smirnov test, making the statistical analyses non-parametric. Therefore, the interquartile mean (IQM) was used as a measure of central tendency [48], and the 25th and 75th ranks and percentiles (Q1 and Q3, respectively) were used as measures of dispersion. The descriptive statistics were performed using Excel version 2101 (Microsoft 365, Redmond, WA, USA). For the inferential statistics, the Kruskal–Wallis test was used to analyze the histopathologic differences between groups, with a post hoc analysis using the Mann–Whitney *U* test. The comparison of serum creatinine and urea levels was analyzed using ANOVA, with a Bonferroni post hoc test. The statistical tests were performed using IBM SPSS version 20 software (IBM SPSS, Chicago, IL, USA), with statistical significance set at *p* < 0.05.

## 3. Results

Significant differences were found in tubular injury (*p* < 0.001), interstitial infiltrate of inflammatory cells (*p* < 0.001), and interstitial fibrosis (*p* < 0.001), when the intergroup analysis (Kruskal–Wallis tests) was carried out. However, no difference was observed in tubular atrophy (*p* = 0.400).

The post hoc analysis was made with the Mann–Whitney *U* test to explore each pair (Table 1). The control group showed normal kidney tissue with no tubular injury or interstitial fibrosis. Gentamicin produced significant tubular injury and interstitial fibrosis (GM + placebo-after placebo) compared with the group without gentamicin (*p* < 0.001, for both analyses), confirming the validity of the animal model (Table 1 and Figure 2).

The group in which neutral electrolyzed saline was administered concomitantly with gentamicin had significantly less tubular injury and interstitial fibrosis than the gentamicin plus placebo group (GM + placebo-after placebo) (*p* < 0.001, for all groups). The comparison of the GM + placebo-after SES vs. GM + placebo-after placebo groups showed that there was no beneficial effect of neutral electrolyzed saline when administered after the end of the gentamicin administration period (*p* = 0.104 for tubular injury, and *p*= 0.872 for interstitial fibrosis) (Figure 2).

There was an increase in the interstitial infiltrate of inflammatory cells in all groups that received gentamicin, regardless of whether SES was administered, compared with the group without gentamicin (Table 1 and Figure 2). It is striking that there were more inflammatory cells in the groups with concomitant administration of neutral electrolyzed saline and gentamicin (GM + SES-after placebo *p* = 0.017, and GM + SES-after SES *p* = 0.036) compared with the group that received gentamicin and placebo (GM + placebo-after placebo) (Table 1 and Figure 2). The total percentage of histopathologically altered renal cortex tissue showed that the concomitant administration of SES and GM was capable of significantly preventing kidney damage (Table 1).

Serum IL-6 values were determined at the end of the study in the groups: (1) Placebo-after placebo (IQM = 14.9 pg/mL, Q1–Q3 = 7.6–23.0); (2) GM + placebo-after placebo (IQM = 5.6 pg/mL, Q1–Q3 = 4.9–7.3); and (3) GM + SES-after placebo (IQM = 28.6 pg/mL, Q1–Q3 = 9.9–41.2). Using the Mann–Whitney *U* test, differences in the groups were determined, and it was found that IL-6 levels were significantly lower in the group that only received GM compared with the group that received GM + SES concomitantly (*p* = 0.028) and compared with the placebo-only group (*p* = 0.046). Meanwhile, the group that received GM + SES showed no significant differences from the group that received only placebo (*p* = 0.173). Table 2 shows a comparison of the serum values for creatinine and urea. In an intergroup analysis via ANOVA, only creatinine levels were different between groups. In the post hoc analysis, creatinine was significantly elevated in the GM + placebo-after placebo group compared with the GM + SES-after placebo (*p* = 0.004), GM + SES-after SES (*p* = 0.006), and GM + placebo-after SES groups (*p* = 0.025), with no differences between the rest of the groups (*p* > 0.05 for the rest of the comparisons). The placebo-after placebo group showed no statistically significant differences with respect to any of the groups (*p* > 0.05 for all comparisons).

## 4. Discussion

Gentamicin administration in an animal model generated tubular injury, interstitial infiltration of inflammatory cells, and renal interstitial fibrosis. The tubular injury and interstitial fibrosis were significantly reduced by the concomitant administration of neutral electrolyzed saline (SES), but there was no beneficial effect when SES was administered after gentamicin application. With the concomitant administration of SES and GM, low serum creatinine levels were also maintained compared with the group that received GM alone. In conclusion, neutral electrolyzed saline mainly had a prophylactic (nephroprotective) effect rather than a therapeutic (reversible) effect on gentamicin nephrotoxicity. The only beneficial effect of administering SES in a period after GM administration was the maintenance of serum creatinine levels that were significantly lower than those of the group that received GM alone.

The gentamicin nephrotoxicity model of the present study generated renal histologic alterations as previously reported [8,9,45,49,50]; however, unlike other investigations, no tubular atrophy was found [45,51]. In addition, doses of gentamicin similar to those in other investigations were administered in our model, but they were applied for 30 days—a period longer than that used in most previous studies [52,53]. Gentamicin nephrotoxicity models differ among studies in terms of dose (80–150 mg/k/day), administration route (subcutaneous or intraperitoneal), and administration time (ranging from 4 to 21 days) [54,55,56,57,58,59]. In the present study, gentamicin doses of 100 mg/k/day were used intraperitoneally for 30 days plus the administration of a high-fat diet to increase the renal and systemic inflammatory environment [30,31,32]. Tubular injury, tubular atrophy, interstitial inflammatory cell infiltrate, and interstitial fibrosis were reported quantitatively as the percentage of the total area of the renal cortex occupied by each lesion. Other studies have used semiquantitative scales that consist of the previous percentage evaluation of histologic lesions and subsequent reports in stages [60,61]. Therefore, the use of a model with a more favorable environment for kidney damage and more precise histologic measurements of kidney damage than those that have featured in previous studies [62,63] can be considered a strength of our investigation. However, the damage we observed was not very extensive, which is an aspect to be considered in future research.

Alternative nephrotoxicity animal models, such as those related to chemotherapeutic drugs like cisplatin, allow clear histopathological damages and have also been studied to evaluate the nephroprotective effect of electrolyzed saline. Oral administration of this substance to cisplatin-induced-renal-injury mice showed a significant nephroprotective effect due to the inhibition of lipidic peroxidation and an increment in antioxidant defense activity. The histopathological analysis of intoxicated mice without electrolyzed saline treatment showed moderate hydropic changes and extensive injury to tubular cells, while histopathological changes in the mice treated with electrolyzed saline revealed lower levels of edema and trace levels of tubular injury [64].

In all cases, the results on nephroprotection due to neutral electrolyzed saline are consistent with previous findings which showed that antioxidant and anti-inflammatory substances can generate a similar effect [65,66]. Inflammation and the exacerbated production of free radicals are known processes involved in the pathogenesis of gentamicin nephrotoxicity [14,67]. The production of proinflammatory cytokines leads to renal structural deterioration, including deterioration due to tumor necrosis factor-alpha (TNF-α), which is involved in the induction of loss of tubular cells secondary to tubular necrosis [54], and IL-6, levels of which increase after induction by GM [68]. The above takes place within the acute period of kidney damage. In addition, gentamicin increases transforming growth factor-β (TGF-β) levels in the renal cortex [36] and serum malondialdehyde levels [69].

Crocin has previously been shown to have protective effects against gentamicin-induced nephrotoxicity in rats; it has been postulated that this is due to its antioxidant and anti-inflammatory properties [45]. Similarly, molecular hydrogen is a potent anti-inflammatory [70] and antioxidant [71] which prevents the low intrarenal oxygenation induced by gentamicin [23]. Additionally, the administration of hydrogen-rich saline has an antifibrotic effect on the skin (on sclerodermatous skin lesions) [72], lungs [17], and liver and decreases circulating TNF-α [70]. All these findings are consistent with the results of the present investigation.

A possible explanation of the nephroprotective effects of neutral electrolyzed saline could be its ability to reduce inflammation [18]. Recently, a preclinical study showed that administering neutral electrolyzed saline in a rheumatoid arthritis model reduced IL-6 levels in a dose-dependent manner [73]. Additionally, molecular hydrogen, which is a component of neutral electrolyzed saline, has previously been reported to reduce levels of malondialdehyde (a marker of lipid peroxidation) and TGF- β1 (a profibrotic cytokine) [22], which would generate a nephroprotective and antifibrotic effect [61,74,75].

Nevertheless, the role of IL-6 in the evolution of kidney damage, determined through its serum levels, seems to be different in the acute and recovery phases. Elevated IL-6 serum levels during an acute process of renal damage are associated with worse renal function and/or pathohistologic changes in animal and human models [68,76], but the role of IL-6 in renal function recovery appears to be very different. Several animal studies suggest that IL-6 regulates antioxidant factors and modifies oxidative stress to protect the kidneys. Even in humans, a higher IL-6 level has been associated with a higher rate of complete renal recovery in survivors with acute kidney injuries admitted to intensive care units [76]. The present study found that IL-6 levels were significantly lower in the animals that received GM alone than in the placebo-only (*p* = 0.046) or gentamicin + SES (*p* = 0.028) groups. Low levels of IL-6 in the group with the greatest kidney damage is consistent with the idea that IL-6 is essential to the recovery process of that organ [76]. The group with concomitant administration of GM + SES did not show this decrease in serum IL-6, and although its values were higher than those in the group that received only placebo (healthy animals), the differences were not significant.

It is important to note that the tested dose of neutral electrolyzed saline could not reverse kidney damage when a subsequent toxic agent was administered. The reversal of already established interstitial fibrosis is documented in reports using other therapies; however, in those studies, gentamicin was administered at a dose of 80 mg/kg/day for 8–10 days [12,77], whereas in our study it was administered for 30 days.

Also noteworthy is the high number of inflammatory cells in the kidney tissues of the mice with fewer histologic lesions due to the nephroprotective effect of neutral electrolyzed saline—a result consistent with the findings of previous studies. The simultaneous administration of L-arginine with gentamicin was reported to contribute to the absence of tubular necrosis and mononuclear cell infiltration associated with tubular regeneration [78]. Carvacrol demonstrated a nephroprotective effect by reducing tubular necrosis without reducing interstitial leukocyte infiltration [79]. Cucumis melo seed extract prevented tubular necrosis, and, depending on the dose, the degree of interstitial infiltration of inflammatory cells was maintained or decreased [16]. These results suggest that the role of the interstitial infiltrate of inflammatory cells in the groups with neutral electrolyzed saline may have been due to renal repair mechanisms rather than deleterious effects, but such a hypothesis needs to be tested in future research. Studies have also shown a reduction in the renal inflammatory infiltrate related to [45,54,80] a decrease in tubular necrosis, interstitial fibrosis, and other histologic lesions of interest. There is a fine line between the component of the interstitial infiltrate induced by nephrotoxicity and inflammation, the fibrotic response, and its possible reparative effect [81,82]. Therefore, studies on this topic and the influence of neutral electrolyzed saline are necessary.

In our research, the mice were fed a high-fat diet (HFD) that contributed to inflammation and renal interstitial fibrosis coupled with gentamicin-induced nephrotoxicity. HFD increases renal cortical mRNA levels of proinflammatory markers (MIP-1α (macrophage inflammatory protein-1α), TNFα, and IL-6) and profibrotic markers (TGF-β1) and increases renal macrophage infiltration [30]. These data could explain the presence of the interstitial infiltrate of inflammatory cells in all the groups, including the control. Therefore, when interpreting the results of the present study or comparing them with future research, the type of diet given to the animals should be taken into account.

Our study has several limitations. First, varied doses of neutral electrolyzed saline, enabling the evaluation of the minimum effective dose to obtain the nephroprotective effect, were not tested. Second, the cellular phenotype of the interstitial component was not characterized. Therefore, the exact role of the interstitial inflammatory cell infiltrates in the murine model of gentamicin-induced nephrotoxicity was not demonstrated. Third, the mouse model utilized represented early (preclinical) chronic kidney disease, with maximum percentages of interstitial fibrosis or tubular injury of 25%, without elevated serum creatinine or urea levels. Future investigations with models that represent more advanced renal disease are necessary. Other studies should demonstrate how neutral electrolyzed saline protects against toxic agents, such as gentamicin, as well as analyze oxidative stress parameters and molecular/biochemical data.

## 5. Conclusions

Our results show that the concomitant administration of neutral electrolyzed saline significantly reduces the nephrotoxic histologic changes caused by gentamicin. However, its administration after toxic damage does not reverse said renal histopathologic lesions. Future research is necessary to analyze the potential clinical use of neutral electrolyzed saline to prevent kidney damage due to toxic agents.

## Figures and Tables

**Figure 1 medicina-59-00397-f001:**
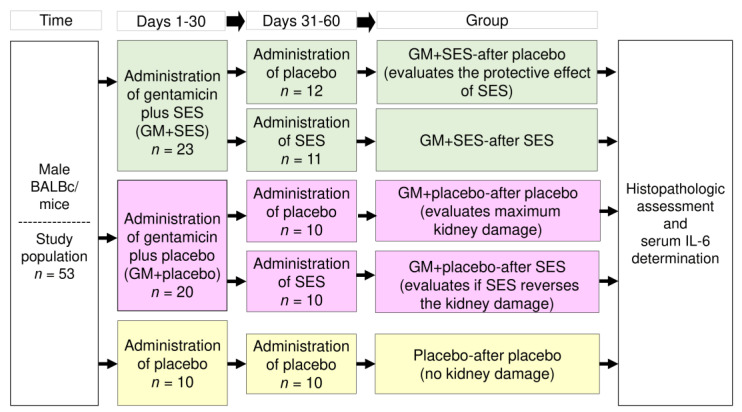
Schematic representation of the study. GM: gentamicin, SES: neutral electrolyzed saline, IL-6: interleukin 6.

**Figure 2 medicina-59-00397-f002:**
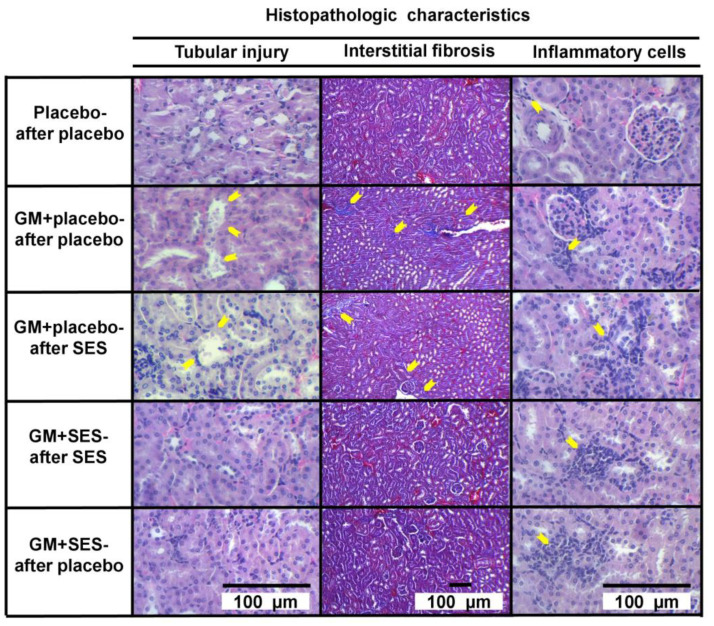
Effect of neutral electrolyzed saline (SES) on histopathologic changes in gentamicin (GM)-induced nephrotoxicity in the Balb/c mouse model. Placebo-after placebo group; GM + placebo-after placebo group; GM + placebo-after SES group; GM + SES-after SES group; and GM + SES-after placebo group. Yellow arrows show areas of tubular injury (flattening of the tubular epithelium, cellular fragments within their lumens, blebbing of the apical membrane into the tubular lumen, and loss of some nuclei), interstitial fibrosis, and interstitial inflammatory cell infiltration in the appropriate section. The GM + SES-after SES and GM + SES-after placebo groups showed less tubular injury and interstitial fibrosis compared with the GM + placebo-after placebo group. In addition, the GM + SES-after placebo and GM + SES-after SES groups had more interstitial inflammatory cell infiltrate than the GM + placebo-after placebo group. Tubular injury and interstitial inflammatory cell infiltrate: hematoxylin–eosin, x400; interstitial fibrosis: Masson’s trichrome, ×100; *n* = 53.

**Table 1 medicina-59-00397-t001:** Paired comparison of kidney histopathologic lesions.

Group	Placebo-after Placebo (*n* = 10)	GM + Placebo-after Placebo (*n* = 10)	GM + Placebo-after SES (*n* = 10)	GM + SES-after SES (*n* = 11)	GM + SES-after Placebo (*n* = 12)
Tubular injury (%)					
IQM	0.00	2.5	0.00	0.00	0.00
Q1–Q3	0–0	0–25	0–0	0–0	0–0
Min–max value *p*-value, comparison	0–0	0–25	0–10	0–0	0–0
vs. placebo alone	NA	0.000 *	0.003 *	1.00	1.00
vs. GM alone	0.000 *	NA	0.104	0.000 *	0.000 *
Interstitial fibrosis (%)					
IQM	0.00	4.00	6.50	0.00	0.00
Q1–Q3	0–0	0–10	0–15	0–0	0–0
Min–max value *p*-value, comparison	0–0	0–25	0–20	0–5	0–0
vs. placebo alone	NA	0.000 *	0.000 *	0.063	1.00
vs. GM alone	0.000 *	NA	0.872	0.000 *	0.000 *
Interstitial infiltrate of inflammatory cells (%)					
IQM	5.50	10	13.50	13.33	10.23
Q1–Q3	5–10	5–15	10–15	8.75–15	10–15
Min–max value *p*-value, comparison	0–10	5–25	5–20	5–20	0–30
vs. placebo alone	NA	0.000 *	0.000 *	0–000 *	0.000 *
vs. GM alone	0.000 *	NA	0.017 *	0.036 *	0.601
	Total percentage of renal cortex tissue altered histopathologically				
IQM	5.5	23.0	21.5	13.3	10.2
Q1–Q3	5–10	5–40	15–30	6.3–18.8	10–15
Min–max value *p*-value, comparison	0–10	5–40	5–40	5–25	5–30
vs. placebo alone	NA	0.000 *	0.000 *	0.000 *	0.000 *
vs. GM alone	0.000 *	NA	0.945	0.009 *	0.005 *

NA: Does not apply, GM: gentamicin, SES: neutral electrolyzed saline, IQM: interquartile mean, Q1: 25th percentile, Q3: 75th percentile, Min–max: smallest and largest values in the data set. (%): Renal histopathologic lesions were calculated as the percentage of the total area observed under the microscope. Total percentage of renal cortex tissue altered histopathologically: result of the sum of percentages of tubular injury, tubular atrophy, interstitial fibrosis, and inflammatory infiltrate in the renal cortex. Comparisons were made with the Mann–Whitney *U* test. * Value of statistical significance: *p* < 0.05.

**Table 2 medicina-59-00397-t002:** Comparison of serum creatinine and urea levels.

Group	Placebo-after Placebo (*n* = 10)	GM + Placebo-after Placebo (*n* = 10)	GM + Placebo-after SES (*n* = 10)	GM + SES- after SES (*n* = 11)	GM + SES- after Placebo (*n* = 12)	*p* ANOVA
Creatinine *	1.17 ± 0.10	1.53 ± 0.56	1.05 ± 0.36	0.98 ± 0.20	0.96 ± 0.23	0.002 *
Urea *	28.34 ± 5.7	32.26 ± 3.61	39.06 ± 17.94	37.0 ± 10.03	31.61 ± 5.51	0.132

* mg/dL. GM: gentamicin, SES: neutral electrolyzed saline. The means + standard deviations are shown. Comparisons were made with ANOVA tests. * Value of statistical significance: *p* < 0.05.

## Data Availability

The current study contains all necessary data. The corresponding author will provide the datasets used and analyzed during the current work upon reasonable request.

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
