# Peer review of "Protective Effect of Neutral Electrolyzed Saline on Gentamicin-Induced Nephrotoxicity: Evaluation of Histopathologic Parameters in a Murine Model"

_medicina, 2023, doi:10.3390/medicina59020397_

Round 1

Reviewer 1 Report

The paper ”Protective effect of neutral electrolyzed saline on gentamicin- 2 induced nephrotoxicity: Evaluation of histopathological param- 3 eters in a murine model” is interesting. Still, it should clarify some ambiguities to improve the quality of the paper.

-I suggest keywords be according to Mesh terms (NCBI) and arrangement according to the English alphabet.

-Is the dosage of SES and the treatment’s duration determined on what basis?

-It needs to show the scale bare of the pathological figure (Figure 2).

-I suggest that tissue damage be scaled and added together in each group and then analyzed statistically (spatially, according to your conclusion, it is more helpful).

-Please rewrite the sentence starting with “The nephroprotective mechanism of neutral electrolyzed saline in reducing fibrosis and tubular …”.

-Please rewrite the sentence starting with “Remembering that the measurement of IL-6 occurred one month after…”.

-Please rewrite the sentence starting with “The nephroprotective effect of interstitial fibrosis and tubular…”.

-Please rewrite the sentence starting with “This finding is different from other researchers who have…”.

-Please rewrite the sentence starting with “Third, in the gentamicin nephrotoxicity model used in this study…”.

-Please check the whole manuscript for punctuation and grammar modification.

Author Response

Comment: The paper ”Protective effect of neutral electrolyzed saline on gentamicin- 2 induced nephrotoxicity: Evaluation of histopathological param- 3 eters in a murine model” is interesting. Still, it should clarify some ambiguities to improve the quality of the paper.

-I suggest keywords be according to Mesh terms (NCBI) and arrangement according to the English alphabet.

Answer: The keywords were rewritten to be a standardized term and identified as MeSH. They were arranged alphabetically

Comment:-Is the dosage of SES and the treatment’s duration determined on what basis?

Answer: Thanks for your observation. The required data was developed and incorporated into the method: “The dose of SES administered per day in the mouse model was equivalent to that given to human patients intravenously in a previous clinical trial, where an 80ml dose of SES per day was shown to have an anti-inflammatory and immunomodulatory effect in patients (doi: 10.3892/etm.2021.10347). Using a calculator that estimates the dose scale between species of different body weights via exponential allometry (http://clymer.altervista.org/minor/allometry.html) https://doi.org/10.1242/jeb.01589, a dose of 150 ul/day for the mouse was found to be a dose equivalent to 80 ml/day in humans. This calculation considered an exponent of 0.80, as well as an average weight of mice and humans of 25gr and 70Kg, respectively. The SES application period was 30 days, being the same duration as the GM application”

Comment:-I suggest that tissue damage be scaled and added together in each group and then analyzed statistically (spatially, according to your conclusion, it is more helpful).

Answer: Excellent suggestion. Now we have added to the results the % of area occupied by some histopathological lesion (sums of all the areas occupied by some lesion). This confirms and allows us to see more clearly the nephroprotective result of the SES. The sum of the histopathological grades of the different types of lesion is also added, also confirming the result.

Comment:-Please rewrite the sentence starting with “The nephroprotective mechanism of neutral electrolyzed saline in reducing fibrosis and tubular …”.

Answer: The sentence was rewritten

Comment:-Please rewrite the sentence starting with “Remembering that the measurement of IL-6 occurred one month after…”.

Answer: The sentence was rewritten

Comment:-Please rewrite the sentence starting with “The nephroprotective effect of interstitial fibrosis and tubular…”.

Answer: The sentence was rewritten

Comment:-Please rewrite the sentence starting with “This finding is different from other researchers who have…”.

Answer: The sentence was rewritten

Comment:-Please rewrite the sentence starting with “Third, in the gentamicin nephrotoxicity model used in this study…”.

Answer: The sentence was rewritten

Comment:-Please check the whole manuscript for punctuation and grammar modification.

Answer: The manuscript has been carefully reviewed by an English-speaking expert.

Reviewer 2 Report

This study evaluates potential prophylactic and/or therapeutic effect of neutral electrolyzed saline in a murine model of gentamicin-induced nephrotoxicity.

There are some misunderstandings regarding the experimental model used in this study. GM-induced model of nephrotoxicity, established in this research, is a model of acute kidney injury. I don t understand why you applied GM, IP for 30 days. There is no analogy with any clinical state.

Also, it is stated in Material and Methods..."mice were fed a high-fat diet to promote a renal and systemic inflammatory environment"... It is known that high-fat diet can provoke metabolic diseases, atherosclerosis, hypertension..., conditions that might interact with GM-induced nephrotoxicity, and make confusion in result interpretation. Maybe, it was easier to establish GM-induced model of nephrotoxicity according to already known quidelines from the literature.

Serum urea and creatinine were not measured to confirm GM-induced nephrotoxicity. Only histopathological findings were presented and serum IL-6 levels. Some results are also confusing, like increase interstitial infiltrate of inflammatory cells observed in the GM+SES group accompanied with decreased tubular necrosis and interstitial fibrosis.

You didn t write what was placebo, you injected instead of SES.

At the end of discussion, limitations of the study were presented. According to my opinion, the main limitations of this study are: First, the established experimental model is not a proper one; Second, as you evaluated the effects of SES (a compound with antiinflammatory, antioxidant and immunomodulatory properties), histopathological findings should be followed by some other analysis (oxidative stress parameters, biochemical analysis..) to complete the results.

Author Response

This study evaluates potential prophylactic and/or therapeutic effect of neutral electrolyzed saline in a murine model of gentamicin-induced nephrotoxicity.

Comment: There are some misunderstandings regarding the experimental model used in this study. GM-induced model of nephrotoxicity, established in this research, is a model of acute kidney injury. I don t understand why you applied GM, IP for 30 days. There is no analogy with any clinical state.

Answer: Thanks for your observation. The animal model is now described more fully in the methodology section. The mouse model analyzed in this study represents an early (preclinical) chronic kidney disease

Comment: Also, it is stated in Material and Methods..."mice were fed a high-fat diet to promote a renal and systemic inflammatory environment"... It is known that high-fat diet can provoke metabolic diseases, atherosclerosis, hypertension..., conditions that might interact with GM-induced nephrotoxicity, and make confusion in result interpretation. Maybe, it was easier to establish GM-induced model of nephrotoxicity according to already known quidelines from the literature.

Answer: Attention is paid to observation in the discussion, as well as its possible influence on the results. When the results and the type of diet are mentioned in the "discussions" section, the following sentence was added: Therefore, when interpreting the results of the present study or comparing them with future research, the type of diet given to the animals should be taken into account

Comment: Serum urea and creatinine were not measured to confirm GM-induced nephrotoxicity. Only histopathological findings were presented and serum IL-6 levels. Some results are also confusing, like increase interstitial infiltrate of inflammatory cells observed in the GM+SES group accompanied with decreased tubular necrosis and interstitial fibrosis.

Answer: A table with serum urea and creatinine results has been included. It has been specified that the model is compatible with an early (preclinical) chronic kidney disease, without elevated serum creatinine or urea. The possible causes of the increase in the inflammatory infiltrate in the renal tissue of the individuals belonging to the groups where nephroprotection was observed have also been discussed.

Comment: You didn t write what was placebo, you injected instead of SES.

Answer: Physiological saline solution for intravenous application was used as placebo. This was described in the methodology

Comment: At the end of discussion, limitations of the study were presented. According to my opinion, the main limitations of this study are: First, the established experimental model is not a proper one; Second, as you evaluated the effects of SES (a compound with antiinflammatory, antioxidant and immunomodulatory properties), histopathological findings should be followed by some other analysis (oxidative stress parameters, biochemical analysis..) to complete the results.

Answer: The limitations of the animal model used have been specified in the discussion, as well as the need for future analysis has been emphasized, as the reviewer has rightly suggested.

Reviewer 3 Report

The authors have conducted interesting research about the beneficial effect of neutral electrolyzed saline administration on gentamicin-induced nephrotoxicity.

Minor recommendation,

I suggest including your statistically significant results in the Abstract;

Figure 1, IL-6: interlude 6.? Or “interleukin -6”;

Row 288, the meaning of the TGF β abbreviation should be introduced;

The Conclusions can be enriched with authors' future perspective regarding this field. 

Author Response

The authors have conducted interesting research about the beneficial effect of neutral electrolyzed saline administration on gentamicin-induced nephrotoxicity.

Minor recommendation,

Comment: I suggest including your statistically significant results in the Abstract;

Answer: Unfortunately we cannot extend the information in the abstract due to the word limit allowed by the journal.

Comment: Figure 1, IL-6: interlude 6.? Or “interleukin -6”;

Answer: Thanks for the observation. The mistake was corrected

Comment: Row 288, the meaning of the TGF β abbreviation should be introduced;

Answer: Thanks for the observation. The mistake was corrected

Comment: The Conclusions can be enriched with authors' future perspective regarding this field.

Answer: Thanks for your observation. Perspectives on future research have been incorporated into the conclusions.

Round 2

Reviewer 2 Report

The authors significantly improved sections Material and Methods and Discussion. I agree to accept paper for publication.

Author Response

Comment: Comments and Suggestions for Authors

The authors significantly improved sections Material and Methods and Discussion. I agree to accept paper for publication.

Answer: Thank you very much for your comment